# P2Y_1_ and P2Y_12_ Receptors Mediate Aggregation of Dog and Cat Platelets: A Comparison to Human Platelets

**DOI:** 10.3390/ijms26031206

**Published:** 2025-01-30

**Authors:** Reece A. Sophocleous, Stephen J. Curtis, Belinda L. Curtis, Lezanne Ooi, Ronald Sluyter

**Affiliations:** 1Molecular Horizons and School of Chemistry and Molecular Bioscience, University of Wollongong, Wollongong, NSW 2522, Australia; reece.sophocleous@sydney.edu.au (R.A.S.); lezanne@uow.edu.au (L.O.); 2Your Village Vet Balgownie, Balgownie, NSW 2519, Australia; stephen.curtis@yourvillagevet.com.au (S.J.C.); belinda.curtis@yourvillagevet.com.au (B.L.C.)

**Keywords:** platelet, P2Y_1_ receptor, P2Y_12_ receptor, purinergic receptor, purinergic signalling, ADP, dog, cat, veterinary medicine, thrombosis, haemostasis, stroke

## Abstract

Thrombosis is one of the most prevalent and serious health issues amongst humans. A key component of thrombotic events is the activation and aggregation of platelets, of which the P2Y_1_ and P2Y_12_ receptors play a crucial role in this process. Despite a breadth of knowledge on thrombosis and its mechanisms and treatment in various disorders in humans, there is less of an understanding of the expression and exact role of these receptors in companion animals such as dogs and cats. Therefore, this study aimed to investigate P2Y_1_ and P2Y_12_ receptors on dog and cat platelets in platelet-rich plasma and compare them to human platelets. Immunoblotting revealed the presence of P2Y_1_ and P2Y_12_ receptor proteins on dog and cat platelets, although relative amounts of each receptor appeared to contrast those of human platelets, with increased amounts of P2Y_1_ compared to P2Y_12_ receptors in dogs and cats. Using a modified 384-well plate aggregation assay, designed for use with small volumes, the human P2Y_1_ and P2Y_12_ receptor agonists adenosine 5′-diphosphate and 2-methylthio-adenosine 5′-diphosphate caused aggregation of dog and cat platelets. This aggregation was near-completely inhibited by the selective P2Y_12_ antagonist ticagrelor. Aggregation of dog and cat platelets was partly inhibited by the human P2Y_1_ receptor antagonist MRS2179. The agonist and antagonist responses in dog and cat platelets were like those of human platelets. In contrast, the aggregation of dog platelets in the absence of added nucleotides was two-fold greater than that of cats and humans. This study indicates that platelets of cats and dogs possess functional P2Y_1_ and P2Y_12_ receptors that can be inhibited by human antagonists. The data presented suggest differing roles or responses of the platelet P2Y receptors in dogs and cats compared to humans but also highlight the potential of using currently available P2Y_1_ or P2Y_12_ antiplatelet drugs such as ticagrelor for the treatment of thrombosis in these companion animals.

## 1. Introduction

Haemostasis and thrombosis play critical roles in human health and disease [1], as well as the health and disease of companion animals such as dogs and cats [2]. A key factor in these processes is the regulation of platelet activation and aggregation, which plays important roles in cardiovascular [3,4,5], renal [6], and hepatic [7] disorders. Current treatments for thrombotic events in humans commonly include selective inhibitors of the platelet P2Y_12_ receptor, such as ticagrelor, cangrelor, clopidogrel, and prasugrel [8], which are among the most prescribed group of drugs in people [9]. In contrast, treatments for thrombotic events in companion animals are primarily limited to heparin and aspirin [10]. Preliminary studies have demonstrated the therapeutic potential of P2Y_12_ inhibitors in dogs [11,12,13,14,15,16,17] and cats [18,19,20], suggesting an important role for the P2Y_12_ receptor in the activation and aggregation of platelets in these companion animals.

In humans, platelets possess P2Y_1_ and P2Y_12_ receptors, which form G_q_ and G_i_ protein-coupled receptors, respectively [21]. These receptors are activated most potently by adenosine 5′-diphosphate (ADP) and its synthetic analogue 2-methylthio-adenosine 5′-diphosphate (2MeSADP) and play important roles in platelet activation and aggregation [22]. In humans, P2Y_1_ receptor activation mediates Ca^2+^ mobilisation from intracellular stores, and initiates platelet aggregation [23,24]. Meanwhile, P2Y_12_ receptor activation inhibits adenylyl cyclase and cyclic adenosine monophosphate production, which amplifies and sustains the platelet aggregation response mediated by the P2Y_1_ receptor [25,26]. The molecular mechanisms linking P2Y receptors with the activation of human platelets have been extensively studied using “gold standard” platelet function tests, such as light transmission aggregometry (LTA) assays, which are a crucial component in haemostatic disorder diagnostics [27].

Compared to human platelets, studies directly investigating the expression and function of P2Y_1_ and P2Y_12_ receptors in dog and cat platelets are limited. Early studies observed differences in ADP-induced aggregation of dog and human platelets and briefly attributed these to differences in purinergic receptors between the species [28], while another study observed ADP-induced aggregation of cat platelets [29]. However, neither study was able to provide a direct link to platelet P2Y receptors. The cloning and molecular characterisation of the human platelet P2Y_1_ [23,24] and P2Y_12_ receptors [30] and later identification of the genes coding these receptors in dogs and cats [31,32] have provided a framework for studies linking genetic evidence of P2Y_1_ or P2Y_12_ receptor expression and platelet function in dogs or cats. Such studies include the identification of a *P2RY12* gene polymorphism in Greater Swiss Mountain Dogs which exhibit impaired platelet responses to ADP [33,34,35] and a *P2RY1* gene polymorphism in cats which alters their response to the P2Y_12_ antagonist clopidogrel [36].

Studies have continued to record ADP-induced aggregation of dog or cat platelets and abnormal platelet function [37], with some studies applying P2Y_1_ or P2Y_12_ antagonists during dog or cat platelet activation in vivo or ex vivo [20,38,39,40]. Additionally, in vitro studies have suggested that both P2Y_1_ and P2Y_12_ receptors are required for ADP-induced aggregation of dog platelets [41], highlighting the importance of these receptors in companion animals. Despite the available evidence, the presence of P2Y receptors on platelets from dogs and cats is often indirectly linked to their function and assumes pharmacological similarity between human and companion animal P2Y receptors. However, the pharmacological profile of P2Y receptors can differ between species, as illustrated by the differential response of human and dog P2Y_11_ receptors to ADP [42,43].

Considering the importance of platelet P2Y receptors in human medicine, but the limited understanding of the expression and specific role of these receptors in platelets of companion animals, the characterisation of P2Y receptors on dog and cat platelets remains an important area of investigation for veterinary medicine. Therefore, this study aimed to investigate the presence of P2Y_1_ and P2Y_12_ on platelets from dogs and cats, in comparison to those of humans, and compare the aggregation of platelets from these species following activation and inhibition of P2Y_1_ or P2Y_12_ receptors.

## 2. Results

### 2.1. Dog, Cat, and Human Platelets Possess P2Y Receptors

Integrin β3/glycoprotein IIIa (CD61) can be used to identify platelets in platelet-rich plasma (PRP) from dogs [44], cats [45], and humans [46]. In this study, PRP isolated from whole blood of dogs, cats, or humans, was labelled with an allophycocyanin (APC)-conjugated anti-CD61 (or isotype) antibody and analysed by flow cytometry to determine the proportion of CD61^+^ platelets. Platelets were gated by forward scatter and side scatter, and this gate was used to identify CD61^+^ platelets (Appendix A). Routine analyses of dog, cat, and human PRP revealed that PRP contained >96% CD61^+^ platelets consistently across all species (Appendix A).

To examine the presence of P2Y_1_ or P2Y_12_ receptors in platelets from dogs, cats, or humans, whole lysates of dog, cat, or human platelets were analysed by immunoblotting using an anti-P2Y_1_ or anti-P2Y_12_ receptor antibody to determine the presence of P2Y_1_ or P2Y_12_ receptors, respectively. Incubation with an anti-P2Y_1_ antibody revealed major protein bands at 53 and 49 kDa in dog platelets and a major protein band at 55 kDa in cat platelets (Figure 1A). A 55 kDa protein was also observed in human platelets, although less distinct (Figure 1A). In contrast, incubation with an anti-P2Y_12_ antibody revealed major bands at 40 kDa and 75 kDa in human platelets, with only minor bands at these molecular weights in cat platelets, and a minor band at 37 kDa in dog platelets (Figure 1B). Together, these data suggest that dog, cat, and human platelets express P2Y_1_ and P2Y_12_ receptor proteins, although to varying degrees, making direct comparisons difficult.

### 2.2. ADP and 2MeSADP Induce Aggregation of Dog, Cat, and Human Platelets in a Concentration-Dependent Manner

Next, a modified LTA assay was used to assess and compare the effects of P2Y_1_ and P2Y_12_ receptors on the aggregation of platelets from dogs, cats, and humans. LTA assays have been used previously to show that ADP and 2MeSADP can induce the aggregation of platelets from dogs, cats, or humans [28,41,47]. However, the use of these assays in studies involving companion animals has been limited, in part due to ethical restrictions on the maximum volumes of blood that can be collected. To account for this, the LTA assay was adapted from a 96-well plate aggregation assay used in a previous study [48] to a 384-well plate assay to allow the study of platelets from small sample volumes. PRP and platelet-poor plasma (PPP) isolated from dogs, cats, or humans were incubated in the absence or presence of increasing concentrations of ADP or 2MeSADP, and platelet aggregation was quantified by measuring absorbance at 595 nm (A_595_) over a 45 min period. At high nucleotide concentrations, near-maximal aggregation was observed within 10–15 min for platelets from dogs, cats, or humans (Appendix A). A 10 min incubation time was selected for comparisons between species.

ADP or 2MeSADP induced a concentration-dependent aggregation of platelets from dogs, cats, or humans (Figure 2A–C). Consistent with previous studies on human P2Y receptors [49], 2MeSADP was significantly more potent than ADP in mediating the aggregation of platelets from dogs (Figure 2A; pEC_50_ 8.50 ± 0.26 vs. 6.05 ± 0.26, *p* < 0.001), cats (Figure 2B; pEC_50_ 8.24 ± 0.16 vs. 6.06 ± 0.16, *p* < 0.001), and humans (Figure 2C; pEC_50_ 8.02 ± 0.20 vs. 6.02 ± 0.09, *p* < 0.001). Despite differences in aggregation kinetics between dog, cat, and human platelets (Appendix A), there were no significant differences in the potency of ADP or 2MeSADP between any species when analysed by one-way ANOVA.

### 2.3. The P2Y_12_ Receptor Antagonist Ticagrelor Inhibits Aggregation of Dog, Cat, and Human Platelets

Ticagrelor is a selective P2Y_12_ receptor antagonist that has well-established inhibitory effects on nucleotide-induced platelet aggregation in humans [8], but limited pharmacological data regarding its effect on dog and cat platelets. To investigate the effect of ticagrelor on platelets of dogs and cats, in comparison to human platelets, PRP and PPP were pre-incubated with increasing concentrations of ticagrelor then incubated in the absence or presence of ADP or 2MeSADP at the approximate EC_80_ for each species (determined from Figure 2A–C). Ticagrelor inhibited ADP- and 2MeSADP-induced aggregation of platelets from dogs (Figure 3A), cats (Figure 3B), and humans (Figure 3C), with maximal inhibition observed between 1 and 3 µM for dog and cat platelets and between 3 and 10 µM for human platelets. There were no significant differences in the potency of ticagrelor in response to activation by ADP between dog (pIC_50_ 6.04 ± 0.09), cat (pIC_50_ 6.31 ± 0.04), and human (pIC_50_ 6.46 ± 0.29) platelets when analysed by one-way ANOVA. In contrast, the potency of ticagrelor in response to activation by 2MeSADP differed significantly between dog (pIC_50_ 6.54 ± 0.05), cat (pIC_50_ 6.31 ± 0.01), and human (pIC_50_ 6.00 ± 0.04) platelets, as revealed by one-way ANOVA with multiple comparisons between all species (*p* < 0.05). There was also a significant difference in the potency of ticagrelor in response to activation by ADP compared to 2MeSADP for dog platelets (*p* < 0.05), but not for cat or human platelets. Collectively, these data reveal that ticagrelor can inhibit ADP- and 2MeSADP-induced aggregation of dog, cat, and human platelets, with some differing effectiveness between species.

### 2.4. The P2Y_1_ Receptor Antagonist MRS2179 Partly Inhibits Aggregation of Dog, Cat, and Human Platelets

MRS2179 is a selective P2Y_1_ receptor antagonist that can inhibit nucleotide-induced aggregation of human platelets [50]. To further elucidate the role of P2Y_1_ receptors in the aggregation of platelets from dogs, cats, and humans, PRP and PPP were pre-incubated in the absence or presence of increasing concentrations of MRS2179 and then incubated in the absence or presence of ADP or 2MeSADP at the approximate EC_80_ for each species. Pre-incubation with MRS2179 partly inhibited ADP- and 2MeSADP-induced aggregation in platelets from dogs (Figure 4A), cats (Figure 4B), and humans (Figure 4C), with maximal inhibition observed near 30 µM for dog platelets and near 10 µM for cat and human platelets. There were no significant differences in the potency of MRS2179 in response to activation by ADP between dog (pIC_50_ 5.34 ± 0.12), cat (pIC_50_ 5.34 ± 0.24), and human (pIC_50_ 5.51 ± 0.17) platelets when analysed by one-way ANOVA. In contrast, there was a significant difference in the potency of MRS2179 in response to activation by 2MeSADP between dog (pIC_50_ 5.35 ± 0.11) and human (pIC_50_ 6.21 ± 0.20) platelets (*p* < 0.01); however, no significant differences were observed between these species and cat platelets (pIC_50_ 5.77 ± 0.09). There were no significant differences in the potency of MRS2179 in response to activation by ADP compared to 2MeSADP for any species. Of note, MRS2179 appeared to be a more effective inhibitor of ADP-induced aggregation in dog platelets compared to cat and human platelets (Figure 4A–C).

### 2.5. Aggregation of Dog Platelets in the Absence of ADP or 2MeSADP Is Greater than That of Platelets from Cats or Humans

The present study has demonstrated that ADP and 2MeSADP can induce platelet aggregation through the activation of P2Y_12_ receptors and, to a lesser extent, P2Y_1_ receptors. However, in the absence of these nucleotides, the aggregation of dog platelets was consistently greater than that of cat or human platelets, with aggregation continuing long after maximal aggregation was observed in cat or human platelets (Appendix A). Therefore, to further investigate the differences in platelet aggregation between the three species, percent aggregation data were pooled from samples and compared for platelets in the absence or presence of their respective EC_80_ for ADP or 2MeSADP (in the absence of any antagonist) after 10 min and 45 min of aggregation.

At 10 min, the aggregation of dog, cat, or human platelets in the presence of nucleotide was significantly greater than that of corresponding platelets in the absence of nucleotide (Figure 5A,B). At this same time point, only the aggregation of cat platelets in the presence of either ADP or 2MeSADP was significantly greater than that of dog or human platelets in the presence of corresponding nucleotide. No significant difference was observed between dog and human platelets in presence of the corresponding nucleotides at 10 min. There were also no significant differences in platelet aggregation among any of the species in the absence of nucleotides at 10 min (Figure 5A,B).

Consistent with the data at 10 min, the aggregation of dog, cat, or human platelets in the presence of nucleotides was significantly greater than that of the corresponding platelets in the absence of nucleotides at 45 min (Figure 5C,D). At this 45 min time point, a significant difference was also observed for the aggregation of dog platelets in the presence of ADP and cat platelets in the presence of 2MeSADP compared to human platelets in the presence of the corresponding nucleotide (Figure 5C,D); however, these differences were less pronounced than that observed at 10 min. Notably, the aggregation of dog platelets in the absence of nucleotides was two-fold and significantly greater than that of cat and human platelets at 45 min (Figure 5C,D). Collectively, these data suggest that in this 384-well plate LTA assay, cat platelets aggregate at a faster rate than dog or human platelets when in the presence of nucleotides. Conversely, in the absence of nucleotides, dog platelets demonstrate greater percent aggregation compared to those of cats or humans, over a longer period (45 min).

## 3. Discussion

The present study aimed to examine the presence and function of P2Y_1_ and P2Y_12_ receptors on dog and cat platelets and compare them to human platelets. Through immunoblotting analyses, it was demonstrated that dog and cat platelets possess P2Y_1_ and P2Y_12_ receptor proteins; however, differences in these profiles between these species and humans were observed. P2Y_1_ receptor proteins were detected between 49 and 55 kDa across the three species, values slightly greater than the predicted molecular weight of P2Y_1_ (42 kDa) but close to that demonstrated for this antibody (47 kDa) in a separate study on human and mouse platelets [51]. P2Y_12_ receptor proteins were detected between 37 and 40 kDa for all species, which is consistent with the predicted (39 kDa) and previously reported molecular weight of P2Y_12_ receptor proteins in human platelets [52,53]. A 75 kDa P2Y_12_ receptor protein was also detected in cat and human platelets, which is consistent with the detection of larger P2Y_12_ receptor protein products in human platelets in these previous studies. This 75 kDa protein was not observed in dog platelets in the present study.

Despite similarities in the amounts of actin between the species, P2Y_1_ receptor proteins were readily detected in dog and cat platelets, but to a much lesser extent in human platelets. In contrast, P2Y_12_ receptor proteins were readily detected in human platelets, but to a much lesser extent in dog and cat platelets. Differences in protein amounts are not unexpected when comparing species using antibody-based techniques due to species-specific differences in antibody specificity or avidity. The anti-P2Y_1_ receptor antibody (APR-021) recognises the SDEYLRSYFIYSMC peptide, corresponding to residues 207–220 of the human P2Y_1_ receptor. This epitope is identical to that of the available cat P2Y_1_ protein sequence (XP 023116226.1); however, it differs by an isoleucine-to-valine residue substitution (SDEYLRSYFVYSMC) in the dog P2Y_1_ protein sequence (NP_001180602.1). Given the epitope similarity between cat and human P2Y_1_ receptors, it is unlikely that antibody specificity would result in the large difference in relative amounts of this receptor between these species. The anti-P2Y_12_ receptor antibody (APR-020) recognises the CTAENTLFYVKES peptide, corresponding to residues 270–282 of the human P2Y_12_ receptor, which differs by a single threonine-to-serine substitution (CSAENTLFYVKES) in both dog (NP_001003365.1) and cat (XP_011284477.1) P2Y_12_ receptors. Although this represents a minor change, it cannot be ruled out that differences in amounts of P2Y_12_ receptors in dogs or cats, in comparison to humans, are not a result of the antibody specificity.

Despite current challenges in directly comparing amounts of P2Y_1_ and P2Y_12_ receptors between species with antibodies directed towards the human epitopes, platelets in PRP of dogs, cats, and humans all displayed CD61, which has been used previously to identify platelets in these three species [44,45,46]. Furthermore, the known nucleotide agonists of human platelet P2Y receptors, ADP and 2MeSADP, were observed to induce aggregation of platelets from dogs and cats, with respective agonist potencies similar to that observed with platelets from humans. In addition, 2MeSADP was found to be a more potent inducer of dog, cat, and human platelet aggregation than ADP, consistent with a previous observation of human P2Y_1_ and P2Y_12_ receptors [54]. Notably, cat platelets appeared to undergo nucleotide-induced aggregation much more rapidly than dog or human platelets, which is consistent with reports of increased sensitivity of cat platelets to aggregation in vivo [55,56].

The LTA assay is considered to be the “gold standard” for platelet function testing [57] and allows for the assessment of the reversible and irreversible phases of platelet aggregation, but traditional aggregometers are limited by the requirement for larger sample volumes and low sample throughput [58]. The lack of a platelet function test that allows for small-volume analyses with higher-throughput screening has potentially limited the ability to study the ex vivo aggregation of platelets from small companion animals including small dogs and cats, as well as rodents, in experimental settings. Several studies have successfully applied the principle of LTA to multiwell plates for assessing human platelet aggregation [48,59,60,61], providing a viable method for testing platelet function using small sample volumes in a laboratory setting. Consistent with these studies, the data presented in this study have demonstrated that this readily accessible and semi-high-throughput 384-well plate LTA assay can be used to measure aggregation of platelets in small volumes of PRP from dogs, cats, and humans. However, the current study was limited in sample size, with each set of experiments undertaken using three different donors per species. Furthermore, plate-based methods such as that utilised in the present study are limited in their ability to assess platelet shape change and the separate phases of aggregation due to a lack of real-time monitoring. Therefore, the cost of lower data resolution should first be weighed against the benefits such as utilising small sample volumes and a greater ability to test against multiple compounds at different concentrations. Although the use of plate-based LTA assays for assessing dog and cat platelet aggregation is yet to be reported, agonist-induced aggregation curves observed in human platelets throughout this study were comparable with previous studies utilising similar plate-based LTA assays [48,61]. This suggests that these assays provide a reliable measure of platelet aggregation.

The data presented in this study demonstrate that the selective P2Y_12_ inhibitor ticagrelor [62] can inhibit ADP- and 2MeSADP-induced aggregation of dog and cat platelets, with similar potency and effectiveness to that observed in human platelets. These data are consistent with preclinical studies demonstrating that ticagrelor maintains its antiplatelet effects in dogs in vivo [12,16]. Moreover, to the best of our knowledge, these data demonstrate the effect of ticagrelor on nucleotide-induced aggregation of cat platelets for the first time. These data support the blockade of cat platelet aggregation in vivo by another P2Y_12_ receptor antagonist, clopidogrel [63], and provide evidence for the potential use of ticagrelor as a treatment or preventative measure for thrombosis in cats. While the safety and efficacy of ticagrelor use in cats should first be validated, this may provide a more effective alternative to clopidogrel, which is reported to have limited effectiveness in cats [36,64].

The selective P2Y_1_ receptor antagonist MRS2179 has been observed to competitively inhibit the aggregation of human washed platelets, where 10 μM MRS2179 nearly completely inhibited platelet aggregation induced by 3 μM ADP [50]. Similar observations were also reported in dog washed platelets, where 100 μM MRS2179 completely inhibited aggregation induced by 200 nM 2MeSADP [41]. Despite this, MRS2179 only partially inhibited dog, cat, and human platelet aggregation induced by ATP or 2MeSADP in the present study. This difference may be associated with the use of apyrase in washed platelet preparations in these previous studies, which may have reduced the potential desensitisation of P2Y_1_ receptors prior to analyses.

Throughout the current study, it was repeatedly observed that the spontaneous aggregation of dog platelets was much greater and more variable than that observed for cat or human platelets, with dog platelet aggregation continuing to increase at a steady rate over 45 min. This variation in basal data in dog platelets may have contributed to the reduced difference observed between platelet aggregation in the absence and presence of nucleotides. The exact reason for this variation remains unclear; however, it may be a result of a number of factors including age, sex, breed, and neutering status, which are known to have an impact on platelets in dogs [65]. There are a limited number of studies reporting resting variations in dog platelet aggregation. Platelet aggregation has been observed previously in dogs in the absence of agonists after long-term (>20 min) aggregation assays [66], which could be linked to the use of non-sodium citrate anticoagulants [67] or concentrations of sodium citrate different from that used in the current study [68]. Consistent with our observations, high biological variability in platelet function has been reported within and between platelet samples from healthy dogs studied over four weeks [69]. Other studies have analysed ex vivo aggregation of dog or cat platelets exposed to ADP or 2MeSADP using LTA (lumi-aggregometer) [13,28,41,70,71,72], whole blood impedance aggregometry (multiplate analyser) [12,15,66,73,74,75], platelet function analysers (PFA-100/200) [76,77], or a combination of methods [78,79,80,81]. However, given the standardised nature of many of these point-of-care assays, platelet activation in the absence of nucleotides or over a longer period of activation is not generally presented or reported in most prior studies. This suggests a need for further studies with larger sample sizes to provide clarity on the spontaneity and variation of dog platelet activation compared to human or cat platelets. Moreover, notable assay variability exists for platelet function worldwide [57,82,83,84], so it remains to be determined how the assay used in this study compares with other platelet assays. Therefore, the methods used for assessing dog platelets should be carefully selected to limit spontaneous aggregation and variation in data.

In conclusion, this study provides direct evidence of P2Y_1_ and P2Y_12_ receptors on dog and cat platelets and, with comparisons between these and human platelets, demonstrates species-specific differences. Dog and cat platelets could be stimulated to undergo aggregation following incubation with known human P2Y_1_ and P2Y_12_ receptor agonists, ADP and 2MeSADP. This platelet aggregation could be either near-completely inhibited by the selective P2Y_12_ antagonist ticagrelor or partially inhibited by the selective P2Y_1_ receptor antagonist MRS2179. This study highlights the potential for targeting platelet P2Y receptors in companion animals using currently available P2Y_1_ or P2Y_12_ inhibitors for the treatment of thrombosis.

## 4. Materials and Methods

### 4.1. Compounds and Reagents

Buffer reagents, BisTris, phenylmethylsulphonyl fluoride, and dithiothreitol (DTT) were from Amresco (Solon, OH, USA). APC-conjugated mouse anti-human CD61 (cat. No. 564174; clone VI-PL2) and isotype control (cat. No. 554681; clone MOPC-21) monoclonal antibodies were from BD Biosciences (San Diego, CA, USA). PBS (cat. No. P4417), 6-aminohexanoic acid, n-dodecyl β-_D_-maltoside, cOmplete EDTA-free protease inhibitor cocktail, rabbit anti-actin polyclonal antibody (cat. No. A2066), gelatine from bovine skin (cat. No. G9391), ADP (30 mM stock in H_2_O; cat. No. A2754), and dimethyl sulfoxide (DMSO) were from Sigma-Aldrich (St Louis, MO, USA). Mini-Protean TGX Stain-Free gels, Precision Plus Protein Dual Colour standards, and nitrocellulose membrane were from Bio-Rad (Hercules, USA). Rabbit anti-human anti-P2Y_1_ (cat. No. APR-021) and rabbit anti-human anti-P2Y_12_ (cat. No. APR-020) polyclonal antibodies were from Alomone Labs (Jerusalem, Israel). Horseradish peroxidase-conjugated goat anti-rabbit IgG secondary antibody was from Rockland Immunochemicals (Pottstown, PA, USA). SuperSignal West Pico Chemiluminescent Substrate was from ThermoFisher Scientific (Waltham, PA, USA). 2MesADP (10 mM stock in H_2_O; cat. No. 1624) was from Tocris Bioscience (Bristol, UK). MRS2179 (5 mM stock in H_2_O; cat. No. 10011450) and ticagrelor (10 mM stock in DMSO; cat. No. 15425) were from Cayman Chemical (Ann Arbor, MI, USA).

### 4.2. Whole Blood Collection

All blood from human and animal donors was collected and studied in compliance with institutional (University of Wollongong, Wollongong, NSW, Australia) and ARRIVE (CITE) guidelines, with approval by the University of Wollongong Animal and Human Ethics Committees. Peripheral blood from dogs, cats, and humans was collected into 3.2% sodium citrate vacutainer tubes (BD Biosciences). The collection of dog and cat blood was carried out by trained veterinary clinicians from Your Village Vet Balgownie (Balgownie, NSW, Australia), with pet owner consent. The collection of blood from healthy human donors was carried out by trained phlebotomists. Informed consent was obtained from all pet owners and human donors. Animal donors were randomly selected with no exclusion regarding pedigree status, breed, age, or sex.

### 4.3. Isolation of Platelets from Whole Blood

Platelets were isolated from whole blood using differential centrifugation as described [48]. Peripheral blood in vacutainer tubes was centrifuged at 200× *g* at room temperature for 20 min with the brake off. The top layer (PRP) was gently removed from the tube and transferred to a sterile 15 mL tube and placed on a shaker (60 rpm) at room temperature until required. The vacutainer tubes, containing the remaining blood and plasma, were centrifuged at 1000× *g* for 10 min. The upper layer (PPP) was then transferred to a sterile 15 mL tube. Platelet counts were conducted using a haemocytometer (Boeco, Hamburg, Germany) to ensure platelet concentration was within a normal range of 200–400 × 10^6^ platelets.mL^−1^. Samples outside of this range were excluded from this study.

### 4.4. Flow Cytometry

The anti-human CD61 monoclonal antibody (clone VI-PL2) can bind dog and cat CD61 [44,85]. A total of 10 μL of whole blood and PRP was incubated with APC-conjugated anti-CD61 or isotype control antibody at 4 °C for 20 min. Samples were washed once with Tyrode’s buffer (134 mM NaCl, 12 mM NaHCO_3_, 2.9 mM KCl, 0.34 mM Na_2_HPO_4_, 10 mM HEPES, 10 mM glucose, pH 7.4) at 1000× *g* for 5 min. Platelets were resuspended in Tyrode’s buffer. Data were acquired using a BD LSR II Flow Cytometer (using a 660/20 band pass filter to detect APC) or a BD Fortessa-X20 (using a 670/30 band pass filter to detect APC) and analysed using FlowJo software v8.7.1 (TreeStar Inc., Ashland, OR, USA). The relative percentage of CD61^+^ platelets was determined as the difference in the percentage of gated events between anti-CD61 and isotype antibody-labelled samples.

### 4.5. Immunoblotting

P2Y receptor protein expression was detected by immunoblotting as described [86]. Platelets were concentrated from PRP by centrifugation at 1000× *g* for 10 min and washed three times with ice-cold PBS at 1000× *g* for 10 min. Following incubation in complete lysis buffer (50 mM BisTris, 750 mM 6-aminohexanoic acid, 1 mM phenylmethylsulphonyl fluoride, 1% n-dodecyl β-_D_-maltoside, and 1 cOmplete EDTA-free protease inhibitor cocktail tablet per 10 mL of buffer, pH 7.0) at 4 °C for 60 min with gentle agitation, cells were sheared 10 times through an 18-gauge needle and centrifuged at 16,000× *g* for 15 min at 4 °C. Supernatants were collected, and the protein concentration was determined using the Pierce BCA Protein Assay Kit (ThermoFisher Scientific) according to manufacturer’s instructions.

Supernatants were diluted in 1:4 in loading buffer (250 mM Tris pH 7.4, 8% SDS, 40% glycerol, and 100 mM DTT) and heated at 94 °C for 3 min. Proteins (20 µg) in loading buffer were separated by denaturing SDS-PAGE under reducing conditions (10 mM DTT) using 4–20% Mini-Protean TGX Stain-Free gels (Bio-Rad) in SDS-PAGE running buffer (192 mM glycine, 3.5 mM SDS, 25 mM tris(hydroxymethyl)methylamine). Proteins were then transferred onto a nitrocellulose membrane in chilled transfer buffer (192 mM glycine, 20% methanol, 25 mM tris(hydroxymethyl)methylamine). The membrane was washed three times in Tris-buffered saline solution containing Tween-20 (TBST) (20 mM Tris, 500 mM NaCl, 0.1% Tween-20, pH 7.5) and blocked in blocking buffer (TBST containing 5% skim milk powder) for 60 min at room temperature or overnight at 4 °C. The membrane was incubated with anti-P2Y_1_ (diluted 1:250) or P2Y_12_ (diluted 1:250) receptor or anti-actin (diluted 1:2000) antibodies in blocking buffer overnight at 4 °C. The membrane was then washed three times for 5 min with TBST and incubated with horseradish peroxidase-conjugated goat-anti rabbit antibody (diluted 1:5,000) in blocking buffer at room temperature for 60 min. The membrane was washed as above and visualised using a chemiluminescent substrate and an Amersham Imager 600RGB (GE Healthcare Lifesciences, Parramatta, Australia).

### 4.6. Light Transmission Aggregometry (LTA)

Nucleotide-mediated platelet aggregation was assessed using a 384-well plate LTA assay modified from the 96-well *Optimul* assay [48,60]. The assay design was modified, where appropriate, in line with the recommendations for the standardisation of light transmission aggregometry [87]. PRP and PPP (15 µL per well) were then transferred to gelatine-coated (0.75% *w*/*v*) wells of a 384-well plate (Greiner Bio-One, Frickenhausen, Germany) for the assessment of platelet aggregation. PRP or PPP in 384-well plates was incubated in the absence (Tyrode’s buffer) or presence of ADP or 2MesADP (prepared in Tyrode’s buffer) (total well volume of 20 µL) and then incubated in a SPECTROstar Nano microplate reader (BMG Labtech, Ortenberg, Germany) at 37 °C with orbital shaking at 700 rpm for 10 s cycles. Absorbance at 595 nm was measured before each 10 s shaking cycle, continuing for up to 60 cycles. In some experiments, samples were pre-incubated in the absence or presence of the P2Y_1_ receptor antagonist MRS2179 or the P2Y_12_ receptor antagonist ticagrelor for 15 min prior to activation with ADP or 2MeSADP. To ensure pipetting accuracy, all volumes were dispensed using an Integra Viaflo electronic pipette (Integra Biosciences, Zizers, Switzerland) with reverse pipetting capability. In some experiments, aggregation was plotted as the percent of A_595_ in PPP, which refers to the platelet aggregation calculated as the percent change in light transmittance of PRP at any given time relative to PPP, according to the Formula (1):(1)$$\%\;\mathrm{aggregation}=\frac{\max\;\mathrm{PRP}\;A_{595}\;-\;\mathrm{PRPt}}{\max\;\mathrm{PRP}\;A_{595}\;-\;\min\;\mathrm{PPP}\;A_{595}}\times 100,$$
where max PRP A_595_ represents 0% aggregation of a given PRP well, PRP_t_ is the A_595_ of PRP in the well at any given time, and min PPP A_595_ represents 100% aggregation [88].

### 4.7. Data and Statistical Analysis

Data are presented as mean ± SEM. Antagonist concentration–response curves were plotted as the percent of aggregation response in the absence of an inhibitor. EC_50_ and IC_50_ values are expressed as their negative logarithm (pEC_50_/pIC_50_) ± SEM. Multiple comparisons were analysed using a one-way ANOVA with Bonferroni post hoc test, respectively. Multiple comparisons involving two interdependent variables were analysed using a main-effects model two-way ANOVA with Bonferroni post hoc test. Two-way ANOVA analysis included a Kolmogorov–Smirnov test to determine normal distributions, using *p* = 0.05. All data were observed to have normal distribution apart from Figure 5D (*p* = 0.047), suggesting a deviation from normal distribution in this experiment. Data were analysed using GraphPad Prism 10.4.1 (GraphPad Software, San Diego, CA, USA).

## Figures and Tables

**Figure 1 ijms-26-01206-f001:**
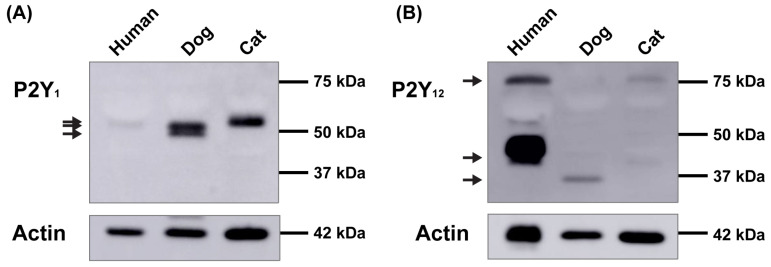
Presence of P2Y_1_ and P2Y_12_ in dog, cat, and human platelets. (**A**,**B**) PRP was isolated from the whole blood of dogs, cats, or humans to obtain platelets. Whole lysates of platelets were analysed by immunoblotting using an (**A**) anti-P2Y_1_ or (**B**) anti-P2Y_12_ receptor antibody (top panels) or (**A**,**B**) anti-actin antibody (bottom panels). Arrows indicate (**A**) P2Y_1_ or (**B**) P2Y_12_ receptors. Results are representative of three independent experiments.

**Figure 2 ijms-26-01206-f002:**
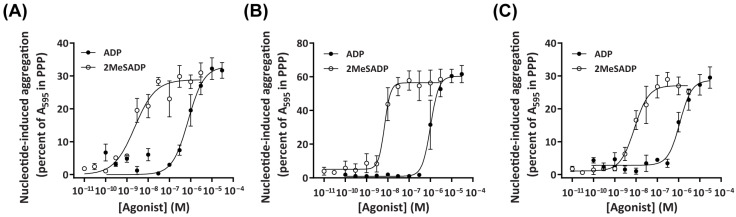
ADP and 2MeSADP activate platelets from dogs, cats, or humans in a concentration-dependent manner. PRP and PPP, isolated from the whole blood of (**A**) dogs, (**B**) cats, or (**C**) humans, were incubated in the absence (Tyrode’s buffer) or presence of ADP or 2MeSADP (as indicated), and platelet aggregation was determined. Concentration–response curves were plotted based on the percent of nucleotide-induced aggregation calculated for each PRP sample, relative to the percent of nucleotide-induced aggregation in the corresponding PPP sample after a 10 min incubation. Data are mean ± SEM from three independent experiments.

**Figure 3 ijms-26-01206-f003:**
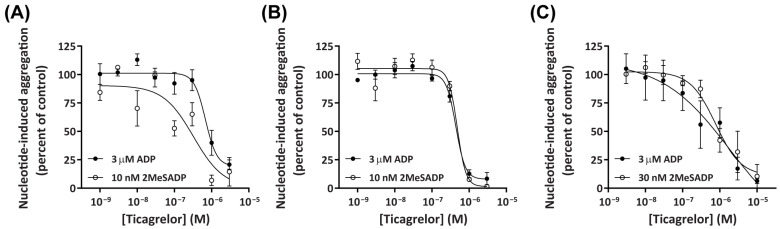
Ticagrelor inhibits ADP- and 2MeSADP-induced aggregation of dog, cat, or human platelets in a concentration-dependent manner. PRP and PPP, isolated from (**A**) dog, (**B**) cat, or (**C**) human whole blood, were pre-incubated in the absence (control; 0.1% DMSO in Tyrode’s buffer) or presence of ticagrelor (as indicated) for 15 min then incubated in the absence (Tyrode’s buffer) or presence of 3 µM ADP or 10–30 nM 2MeSADP (approximate EC_80_ as determined for each species), and platelet aggregation was measured. Concentration–response curves were plotted using nucleotide-induced percent of aggregation calculated for each PRP sample relative to the percent of nucleotide-induced aggregation in the corresponding PPP sample after 10 min incubation and then normalised to the percent of nucleotide-induced aggregation in the absence of ticagrelor. Data shown for each species are mean ± SEM from three independent experiments.

**Figure 4 ijms-26-01206-f004:**
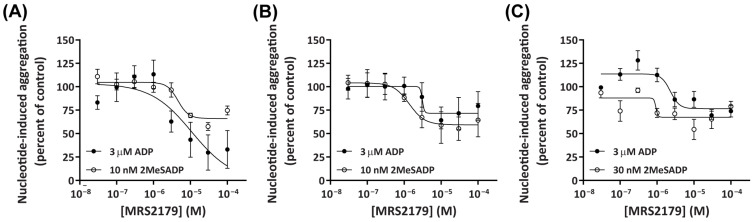
MRS2179 partly inhibits ADP- and 2MeSADP-induced aggregation of dog, cat, or human platelets in a concentration-dependent manner. PRP and PPP, isolated from (**A**) dog, (**B**) cat, or (**C**) human whole blood, were pre-incubated in the absence (control; Tyrode’s buffer) or presence of MRS2179 (as indicated) for 15 min then incubated in the absence (Tyrode’s buffer) or presence of 3 µM ADP or 10–30 nM 2MeSADP (approximate EC_80_ as determined for each species), and platelet aggregation was measured. Concentration–response curves were plotted using nucleotide-induced percent of aggregation calculated for each PRP sample relative to the percent of nucleotide-induced aggregation in the corresponding PPP sample after 10 min incubation and then normalised to the percent of nucleotide-induced aggregation in the absence of MRS2179. Data shown for each species are mean ± SEM from three independent experiments.

**Figure 5 ijms-26-01206-f005:**
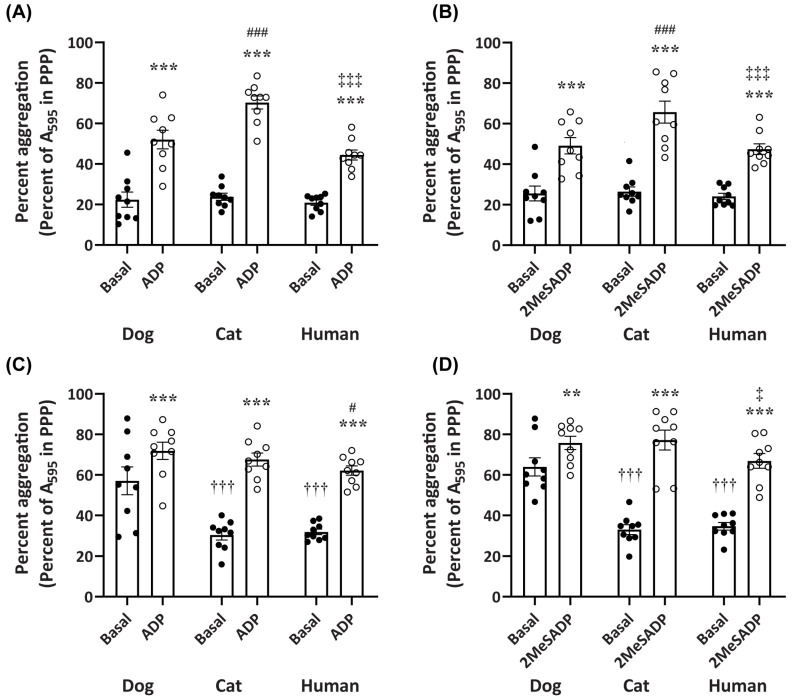
Variation in aggregation between dog, cat, and human platelets. Percent aggregation data in the absence (basal) or presence of (**A**,**C**) 3 µM ADP or (**B**,**D**) 10 nM (dog and cat) or 30 nM (human) 2MeSADP. Data were compared (**A**,**B**) after 10 min of aggregation and (**C**,**D**) after 45 min of aggregation (used throughout to plot concentration–response curves) using a two-way ANOVA with Bonferroni multiple comparisons test; ** *p* < 0.01 and *** *p* < 0.001 compared to corresponding basal, ^†††^
*p* < 0.001 compared to dog basal, ^#^
*p* < 0.05 and ^###^
*p* < 0.001 compared to dog ADP or 2MeSADP, ^‡^
*p* < 0.05 and ^‡‡‡^
*p* < 0.001 compared to cat ADP or 2MeSADP. Data shown for each species are mean percent of platelet aggregation relative to PPP ± SEM from nine independent experiments, where data are derived from samples studied in Figure 2, Figure 3 and Figure 4.

## Data Availability

All data presented in this manuscript are available on request.

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
