# Peer review of "P2Y1 and P2Y12 Receptors Mediate Aggregation of Dog and Cat Platelets: A Comparison to Human Platelets"

_ijms, 2025, doi:10.3390/ijms26031206_

Round 1

Reviewer 1 Report

Comments and Suggestions for Authors

This is quite an interesting and carefully designed study on the function of two types of ADP receptorsv(P2Y1 and P2Y12) in blood platelets of cats, dogs and humans. Shortly after stimulation with ADP, platelets activaton occurs, which is manifested by a platelet shape change, increased  phospholipase C activity, an increase in cytosolic calcium ions, and a suppression of cAMP formation. The Authors demonstrated some differences in the profiles of P2Y receptors beetween comparised species, which is a valuable observation. They also showed some variations in aggregation beetwen dog, cat and human platelets. Unfortunately, they used a multi-well plate-based aggregometry, which makes it dificult to asses platelet shape change or two phases of aggregation induced by ADP.

In many species platelets undergo a primary (reversible) phase of aggregation in response to ADP, whereas in humans, cats and guinea pigs also a secondary phase of (irreverisble) response is present (Packham MA, Rand ML. Historical perspective on ADP-induced platelet activation. Purinergic Signal. 2011;7(3):283-920.) In addition, according the literature most dogs have platelets that undergo only a primary, reversible phase of aggregation in response to ADP. The results in this manuscript do not confirm this, and the description of the procedure for selecting animals or preparing platelets (PRP) raises my concerns. It is worth emphasizing that the presented aggregation results (Figure 5) in dogs between the samples not containing (basal) or stimulated with ADP are not as significantly different as described by the authors in this manuscript. This raises my concerns about the statistical methods used.  In my opinion, p<0.001 in dogs in Figures 5C or 5D is impossible to achieve. Why was a two-tailed Student's t-test used, have Authors ensured that the data distribution is normal (the Kolmogorov-Smirnov test or the Shapiro-Wilk test could be useful)? 

What could be the reason why spontaneous platelet aggregation in dogs was much higher and not much different from that induced by ADP? Did the authors consider that prior exposure of platelets to ADP (e.g. during centrifugation) may cause their desensitization? It would be worth carefully analyzing the results and comparing them with data in the literature. These are interesting studies, but they lack some quality and emphasis the scientific novelity 

Reviewer 2 Report

Comments and Suggestions for Authors

In the paper of Sophocleous and colleagues authors found platelets of cats and dogs possess functional P2Y1 and P2Y12 receptors that can be inhibited by human antagonists. This is a promising conclusion for diagnostics and seems to be a interesting scientific background for further studies. The paper is well organized and the results add important information to the field.

However I have to raise some following comments:

1. Have author considered the reduced sensitivity to antagonists of P2Y1 and P2Y12 receptors, referred to as antagonist resistance, as such phenomena is obseved for aspirin or thienopyridines in human cardiovascular or diabetic patients?

2. Page 2, line 89: there should be "integrin β3" insted of "integrin 3β".

3. Supplementary Figure 1: What for there is presented only gating strategy for platelets identification since authors did show the presence of P2Y receptors on platelets of tested species using flow cytometry? Flow cytometry is better method to identify the presence of P2Y receptors on platelets surface than immunoblotting, in which whole lysate is used. Author did not isolate membrane proteins to analyse the expression of P2Y receptors on platelets using immunobloting. Therefore authors cannot claim that they tested P2Y expression on platelets surface by performing immunoblotin on whole lysates. Authors can only conclude on expression of whole P2Y protein on the surface of platelets and inside platelets.

4. Page 3, lines 102-104: the authors claimed that "A 55 kDa protein was also observed in human platelets, although less distinct, despite loading equal amounts of protein and observing similar amounts of actin between species (Figure 1A)". However I cannot agree with this, since on the Figure 1A the intensity of the actin bands differs between different species, which indicates  loading different amounts of the lysate protein. similar reservations can be seen for the Figure 1B.

5. Did the authors take into account that differences in platelet aggregation between the species studied may result from different platelet counts? The authors do not write  whether they standardized the aggregation measurements to the same number of platelets for the different tested species?

6. Page 4, Figure 2: the Y axis description is not clear - what does it mean "percent of A595 in PPP? How did authors calculate the aggregation?

7. Page 7, Figure 5: some of the aggregation value ranges shown in this figure differ from the aggregation value ranges shown in the figure 2.

8. Did the authors use a combination of both P2Y and P2Y112 receptor inhibitors?

Round 2

Reviewer 2 Report

Comments and Suggestions for Authors

The authors have incorporated the corrections suggested during the revision process and I do not have further comments.